# Non-Coding RNA as a Biomarker in Lung Cancer

**DOI:** 10.3390/ncrna10050050

**Published:** 2024-09-30

**Authors:** Chahat Suri, Shashikant Swarnkar, LVKS Bhaskar, Henu Kumar Verma

**Affiliations:** 1Department of Oncology, Cross Cancer Institute, University of Alberta, Edmonton, AB T6G 1Z2, Canada; csuri@ualberta.ca; 2Department of Biochemistry, C.C.M. Medical College, Bhilai 490020, Chhattisgarh, India; shashikant_bsp@rediffmail.com; 3Department of Zoology, Guru Ghasidas Vishwavidyalaya, Bilaspur 495009, Chhattisgarh, India; lvksbhaskar@gmail.com; 4Department of Immunopathology, Institute of lungs Health and Immunity, Comprehensive Pnemology Center, Helmholtz Zentrum, Neuherberg, 85764 Munich, Germany; 5Lung Health and Immunity, Comprehensive Pneumology Center, Helmholtz Zentrum, Neuherberg, 85764 Munich, Germany

**Keywords:** non-coding RNA, lung cancer, biomarker, therapeutics, mechanism

## Abstract

Introduction: Lung cancer remains one of the most prevalent and deadly cancers globally, with high mortality rates largely due to late-stage diagnosis, aggressive progression, and frequent recurrence. Despite advancements in diagnostic techniques and therapeutic interventions, the overall prognosis for lung cancer patients continues to be dismal. Method: Emerging research has identified non-coding RNAs (ncRNAs), including microRNAs, long non-coding RNAs, and circular RNAs, as critical regulators of gene expression, significantly influencing cancer biology. These ncRNAs play pivotal roles in various aspects of lung cancer pathogenesis, including tumor initiation, progression, metastasis, and resistance to therapy. Results: We provide a comprehensive analysis of the current understanding of ncRNAs in lung cancer, emphasizing their potential as biomarkers for early diagnosis, prognostication, and the prediction of the therapeutic response. We explore the biological functions of ncRNAs, their involvement in key oncogenic pathways, and the molecular mechanisms by which they modulate gene expression and cellular processes in lung cancer. Furthermore, this review highlights recent advances in ncRNA-based diagnostic tools and therapeutic strategies, such as miRNA mimics and inhibitors, lncRNA-targeted therapies, and circRNA-modulating approaches, offering promising avenues for personalized medicine. Conclusion: Finally, we discuss the challenges and future directions in ncRNA research, including the need for large-scale validation studies and the development of efficient delivery systems for ncRNA-based therapies. This review underscores the potential of ncRNAs to revolutionize lung cancer management by providing novel diagnostic and therapeutic options that could improve patient outcomes.

## 1. Introduction

Lung cancer (LC), with its high incidence and mortality rates, poses a significant challenge to global health [1]. LC remains a significant global health concern, with high incidence and mortality rates worldwide. According to the latest estimates from GLOBOCAN 2020, there were approximately 2.2 million new LC cases and 1.8 million lung cancer-related deaths in 2020 [2]. This accounts for about 11.4% of all new cancer cases and 18.0% of all cancer deaths globally. The age-standardized incidence rate (ASIR) for LC was 22.4/100,000 people, while the age-standardized mortality rate (ASMR) was 18.0 per 100,000 [3]. Notably, lung cancer incidence and mortality rates vary significantly across regions, with a 20-fold difference between the highest and lowest rates globally due to drug resistance [4]. Asia bears the largest burden, accounting for over half of all LC cases and 61.9% of mortality worldwide [5]. The majority of lung cancer cases are diagnosed at an advanced stage, leading to limited treatment options and poor survival rates [6]. Hence, there is an urgent need for reliable biomarkers that can facilitate early detection, predict patient outcomes, and monitor therapeutic responses. Non-coding RNAs (ncRNAs), once considered transcriptional noise, have been recognized for their critical roles in regulating gene expression and maintaining cellular homeostasis [7]. ncRNAs, particularly microRNAs (miRNAs), long non-coding RNAs (lncRNAs), and circular RNAs (circRNAs), have been implicated in various cancers, including LC [8]. These molecules exhibit tissue-specific expression patterns and stability in body fluids, making them attractive candidates for non-invasive biomarkers [9,10]. One type of ncRNAs, called lncRNAs, is a class of RNA molecules that exceed 200 nucleotides in length and do not encode proteins [11]. Initially considered mere transcriptional noise, lncRNAs have become crucial regulators of gene expression and cellular processes [12]. In this context, ncRNAs have garnered attention for their potential as both prognostic and diagnostic biomarkers [13]. Their stability in circulation, coupled with their tissue-specific expression patterns, positions them as ideal candidates for non-invasive diagnostic tools. ncRNAs often exhibit tissue-specific expression, which can be leveraged to distinguish between cancerous and non-cancerous tissues [14]. This specificity enhances their utility as biomarkers, as certain ncRNAs may be upregulated or downregulated in LC compared to healthy lung tissue [15].

Recent studies have highlighted the potential of ncRNAs to serve as indicators of tumor presence and progression [16]. Elevated levels of specific ncRNAs in the circulation of LC patients have been associated with advanced disease stages and metastasis [17]. This correlation suggests that ncRNAs could be used not only for initial diagnosis but also for monitoring disease progression and therapeutic response [18]. The diverse roles of ncRNAs in LC can be categorized into several functional types, including oncogenic ncRNAs, which promote tumorigenesis, and tumor-suppressive ncRNAs, which inhibit cancer progression [19]. Emerging research has also highlighted the involvement of ncRNAs in various aspects of LC biology, including the regulation of the epithelial–mesenchymal transition (EMT), a critical process in metastasis [20]. ncRNAs can modulate signaling pathways and interact with proteins to influence cellular behavior, making them integral to understanding the molecular mechanisms underlying LC progression [21]. Furthermore, the advent of high-throughput sequencing technologies has facilitated the identification of ncRNA expression profiles associated with different LC subtypes, paving the way for personalized medicine approaches [22]. Their unique properties as stable, tissue-specific molecules with prognostic and diagnostic potential position them as valuable assets in the fight against LC [23]. As our understanding of ncRNA biology deepens, it is anticipated that these molecules will play an increasingly pivotal role in developing innovative diagnostic tools and targeted therapies, ultimately improving patient outcomes in LC management.

The exploration of ncRNAs in LC has evolved significantly over the past two decades, marking a transformative journey in cancer research [9]. Beginning in the early 2000s, the scientific community recognized that a substantial portion of the genome is transcribed into RNA that does not code for proteins, laying the groundwork for understanding the functional roles of ncRNAs [24]. The discovery of miRNAs in 2006 as key regulators of gene expression catalyzed interest in their implications for cancer, including LC [25]. As research progressed through the 2010s, lncRNAs and circRNAs emerged as critical players in cancer biology, influencing processes such as proliferation, migration, and invasion [26]. The recognition of exosomal ncRNAs in the mid-2010s further underscored their role in intercellular communication and tumor microenvironments [27]. By the late 2010s, specific exosomal lncRNAs and circRNAs were identified as potential diagnostic and prognostic biomarkers for LC, revealing their ability to differentiate between disease stages and contribute to therapeutic resistance [28]. The current decade has seen a surge in research focused on the functional characterization of these molecules, integrating multi-omics approaches to unravel their complex networks in tumorigenesis and metastasis [29]. Recent reviews have emphasized the clinical relevance of exosomal lncRNAs and circRNAs, advocating for their incorporation into precision medicine and personalized treatment strategies, thus highlighting their promise in improving LC management [30]. To better understand the discovery and research history of ncRNAs in LC, it is helpful to review the timeline of ncRNAs (Figure 1).

This timeline shows the significant milestones in ncRNA research related to LC, from the discovery of key ncRNAs to the development of ncRNA-based diagnostic and therapeutic strategies.

## 2. Biological Functions of ncRNAs in Lung Cancer

ncRNAs participate in a wide range of biological processes and pathways that are crucial for LC development and progression. lncRNAs play a pivotal role in regulating gene expression at multiple levels [31]. They can recruit chromatin-modifying complexes to specific genomic regions, influencing the transcriptional landscape of cells. This recruitment can lead to either activating or repressing target genes, depending on the context and the specific lncRNA involved, as illustrated in Figure 2 [32]. lncRNAs can facilitate the formation of transcriptional complexes that enhance the expression of oncogenes or suppress tumor suppressor genes, thereby contributing to cancer progression [33]. One of the most well-studied functions of lncRNAs in cancer is their role as ceRNAs. lncRNAs can bind to miRNAs, effectively “sponging” them and preventing them from interacting with their target mRNAs [34]. This interaction can lead to the increased expression of genes that miRNAs would otherwise downregulate. lncRNA LC-associated transcript 1 (LCAT1) has been shown to sponge miR-4715-5p, leading to the increased expression of the Rac family small GTPase 1 (RAC1), which promotes tumor metastasis [35].

Similarly, MALAT1 has been implicated in sponging miRNAs to enhance the expression of metastasis-related genes [36]. lncRNAs can directly interact with transcription factors, influencing their activity. They can either enhance or inhibit the transcription of target genes by modulating the binding of these factors to DNA. This regulatory mechanism is crucial in cancer, where the dysregulation of transcription factors can lead to uncontrolled cell proliferation and survival [37]. Beyond transcription, lncRNAs also play significant roles in post-transcriptional regulation. They can interact with RNA-binding proteins to influence mRNA splicing, stability, and translation [38]. This interaction can lead to the production of different protein isoforms or the degradation of mRNAs, thereby affecting the overall protein output of the cell. lncRNAs can modulate the stability of mRNAs involved in cell cycle regulation and apoptosis, which are critical processes in cancer [39]. lncRNAs can also affect the stability of proteins by regulating their degradation through the proteasome pathway [40]. By influencing the degradation of specific proteins, lncRNAs can modulate various cellular processes, including cell survival, proliferation, and apoptosis. This function is particularly relevant in cancer, where the stability of oncogenic or tumor suppressor proteins can significantly impact tumor behavior [41]. lncRNAs are involved in the regulation of several key signaling pathways that are critical for cancer progression. They can modulate pathways related to angiogenesis (the formation of new blood vessels), autophagy (the process of cellular degradation and recycling), and immune escape (the ability of cancer cells to evade the immune system). lncRNAs can influence the TGF-β signaling pathway, which is known to play a role in the EMT and metastasis [42]. lncRNAs contribute to the structural organization of the nucleus. They are involved in the formation and maintenance of nuclear compartments, such as paraspeckles, which are important for the regulation of gene expression and RNA processing [43]. By influencing nuclear architecture, lncRNAs can affect the spatial organization of chromatin and transcriptional activity, thereby impacting cellular function. The diverse functions of ncRNAs, particularly lncRNAs, underscore their significance in the molecular mechanisms underlying LC metastasis and progression [44]. Their ability to regulate gene expression, interact with miRNAs, modulate signaling pathways, and influence nuclear architecture highlights their potential as therapeutic targets. Understanding these roles can pave the way for innovative strategies in LC treatment, potentially leading to more effective and personalized therapies. Future research should continue to elucidate the complex regulatory networks involving lncRNAs to harness their potential in clinical applications.

### 2.1. MicroRNAs (miRNAs)

miRNAs are small, single-stranded RNAs that regulate gene expression post-transcriptionally by binding to complementary sequences in target messenger RNAs (mRNAs), leading to mRNA degradation or translation inhibition. In LC, miRNAs can function as oncogenes or tumor suppressors [45]. miRNAs are small, approximately 22-nucleotide-long, single-stranded RNA molecules that play a crucial role in the regulation of gene expression. They are involved in various biological processes, including development, differentiation, proliferation, and apoptosis [46]. miRNAs exert their regulatory effects primarily at the post-transcriptional level by binding to complementary sequences in target mRNAs. This binding can lead to two main outcomes. When a miRNA binds to its target mRNA with perfect or near-perfect complementarity, it can recruit the RNA-induced silencing complex (RISC) to degrade the mRNA, effectively reducing the levels of the corresponding protein [47]. If the complementarity is imperfect, the miRNA may inhibit the translation of the target mRNA without causing its degradation. This results in a decrease in protein synthesis from that mRNA [48]. In the context of LC, miRNAs can function as either oncogenes or tumor suppressors, depending on their expression levels and the specific targets they regulate. Their dysregulation is often associated with cancer initiation, progression, and metastasis. Oncogenic miRNAs, often referred to as “oncomiRs”, are typically overexpressed in cancerous tissues and promote tumorigenesis by targeting tumor suppressor genes. One of the most studied oncomiRs in LC, miR-21, is frequently found to be overexpressed in various types of LC, including non-small-cell LC (NSCLC) [49]. It promotes tumor growth and metastasis by targeting and downregulating several key tumor suppressor genes, as PTEN is a critical regulator of the PI3K/AKT signaling pathway, which is involved in cell survival and proliferation. Targeting PTEN would enhance the activation of this pathway via miR-21, leading to increased cell growth and survival [50]. PDCD4 is a tumor suppressor that inhibits translation and promotes apoptosis. The downregulation of PDCD4 by miR-21 contributes to enhanced cell proliferation and resistance to apoptosis, further facilitating tumor progression [51]. Conversely, some miRNAs act as tumor suppressors by being downregulated in cancer. Their loss of function can lead to the overexpression of oncogenes and contribute to tumorigenesis. This miRNA is often downregulated in LC and is known to target several oncogenes, including those involved in cell cycle regulation and apoptosis [52]. The loss of miR-34 expression can lead to uncontrolled cell proliferation and resistance to cell death, promoting cancer progression [53].

miR-34 is often found to be downregulated in LC tissues compared to normal lung tissues. This downregulation is frequently due to genetic and epigenetic alterations, such as promoter methylation or the loss of genomic loci containing the miR-34 gene [54]. The reduction in miR-34 levels leads to the derepression of its target oncogenes, contributing to the malignant phenotype of LC cells. miR-34 targets multiple oncogenes involved in cell cycle progression, including Cyclin D1, CDK4/6, and E2F family members. These targets are crucial for the G1/S transition in the cell cycle. By downregulating these proteins, miR-34 induces cell cycle arrest, preventing uncontrolled cell proliferation. miR-34 also targets anti-apoptotic proteins such as BCL2 [55]. The loss of miR-34 leads to increased levels of BCL2, which inhibits programmed cell death (apoptosis) and allows cancer cells to survive under conditions that would normally induce cell death. miR-34 can inhibit the EMT, a process by which epithelial cells acquire mesenchymal characteristics, enhancing their migratory and invasive capabilities. Targets of miR-34 in this pathway include SNAIL, SLUG, and ZEB1/2, which are transcription factors that promote the EMT and metastasis [56]. The downregulation of miR-34 removes inhibitory control over cell cycle progression, leading to unchecked cell division and tumor growth. Elevated levels of anti-apoptotic proteins due to miR-34 downregulation contribute to the survival of cancer cells, even in the presence of chemotherapeutic agents or other stressors [57]. By promoting the EMT, the loss of miR-34 facilitates the dissemination of cancer cells from the primary tumor to distant organs, leading to metastasis and worsening patient prognosis [58]. Restoring miR-34 expression in LC cells using miR-34 mimics or gene therapy approaches can inhibit tumor growth and sensitize cancer cells to chemotherapy and radiation. Clinical trials are investigating the use of miR-34-based therapeutics in various cancers, including LC [53]. Combining miR-34 mimics with existing treatments, such as EGFR-TKIs or immune checkpoint inhibitors, may enhance therapeutic efficacy and overcome resistance mechanisms. The mechanisms by which miRNAs exert their effects in LC are multifaceted. miRNAs can modulate key signaling pathways involved in cancer, such as the PI3K/AKT, MAPK, and TGF-β pathways [59]. By targeting components of these pathways, miRNAs can influence processes like cell proliferation, migration, and invasion. MiRNAs can also affect the tumor microenvironment by regulating the expression of cytokines, chemokines, and other factors that influence immune responses and angiogenesis [60]. miR-21 has been implicated in promoting an immunosuppressive environment that favors tumor growth. Some miRNAs are involved in the regulation of the EMT, a process that allows cancer cells to acquire migratory and invasive properties. miR-200 family members are known to suppress the EMT and maintain epithelial characteristics, and their downregulation can facilitate metastasis [61]. The dysregulation of miRNAs in LC presents potential opportunities for therapeutic intervention. Strategies to restore the expression of tumor suppressor miRNAs or inhibit oncogenic miRNAs are being explored as potential therapeutic approaches. Delivering miRNA mimics or inhibitors (antagomiRs) could help restore normal regulatory networks disrupted in cancer [62]. MiRNAs have the potential to serve as biomarkers for LC diagnosis, prognosis, and treatment response. Their stable presence in body fluids, such as blood and sputum, makes them attractive candidates for non-invasive diagnostic tests [45]. Their ability to function as oncogenes or tumor suppressors, along with their involvement in various cellular processes, underscores their importance in cancer progression and metastasis. The specific roles of different miRNAs in LC can provide valuable insights into potential therapeutic strategies and improve patient outcomes.

### 2.2. Long Non-Coding RNAs (lncRNAs)

lncRNAs are transcripts longer than 200 nucleotides that do not encode proteins but can modulate gene expression through various mechanisms, including chromatin remodeling, transcriptional regulation, and post-transcriptional processing. lncRNA MALAT1 is highly expressed in LC and is associated with poor prognosis [63]. It promotes cell proliferation and metastasis by interacting with key regulatory proteins and modulating gene expression. Two of the already known LC biomarkers, circFARSA and circFoxo3, are currently being understood more for the management of LC. CircFARSA has been found to be significantly elevated in the plasma of patients with NSCLC [64]. Its increased expression is associated with tumor aggressiveness and poor prognosis. CircFARSA shows potential as a non-invasive diagnostic and prognostic biomarker for NSCLC, allowing for early detection and monitoring of disease progression through simple blood tests [65].

CircFoxo3, on the other hand, exhibits variable expression patterns across different cancer types, including LC. In NSCLC specifically, circFoxo3 has been reported to be downregulated in tumor tissues and cell lines [66]. It acts as a tumor suppressor by inhibiting cell proliferation, migration, and invasion through various molecular mechanisms, including sponging oncogenic miRNAs. The decreased expression of circFoxo3 in NSCLC patients′ samples correlates with an advanced tumor stage and poorer survival outcomes, highlighting its potential as a prognostic biomarker [67]. Both circFARSA and circFoxo3 demonstrate the clinical utility of circular RNAs as stable, specific biomarkers that can be detected in liquid biopsies. Their altered expression patterns in LC patients provide valuable diagnostic and prognostic information, potentially improving early detection rates and treatment strategies for this deadly disease [68].

circRNAs are a class of ncRNAs characterized by a covalently closed loop structure, which provides them with stability [69]. They can act as miRNA sponges, sequestering miRNAs and preventing them from binding to their target mRNAs. circRNA CDR1 sequesters miR-7, leading to the increased expression of miR-7 target genes that promote LC cell proliferation and invasion [70]. lncRNAs are defined as RNA molecules longer than 200 nucleotides that do not code for proteins. Despite their lack of protein-coding ability, they play crucial roles in regulating gene expression through several mechanisms. lncRNAs can influence the structure of chromatin, which is the complex of DNA and proteins that forms chromosomes [71]. lncRNAs can promote or inhibit the accessibility of certain genes, thereby regulating their expression by interacting with chromatin-modifying complexes. lncRNAs can interact with transcription factors and other regulatory proteins to modulate the transcription of target genes [72]. They can act as scaffolds, bringing together various components of the transcription machinery to enhance or repress gene expression.

lncRNAs can also influence the processing of mRNA, including splicing, stability, and translation. This can affect the levels of proteins produced from those mRNAs. MALAT1 (Metastasis-Associated Lung Adenocarcinoma Transcript 1) is one of the most studied lncRNAs in LC [73]. It is often found to be overexpressed in LC tissues and is associated with a poor patient prognosis. MALAT1 promotes cell proliferation and metastasis through several mechanisms. It interacts with key regulatory proteins involved in cell cycle progression and metastasis. It modulates the expression of genes that are critical for tumor growth and spread, thereby enhancing the aggressive characteristics of LC cells.

### 2.3. Circular RNAs (CircRNAs)

CircRNAs are a unique class of non-coding RNAs characterized by their covalently closed loop structure. This circular formation makes them more stable than linear RNAs, allowing them to persist longer in the cellular environment. CircRNAs can perform various functions; one of the most well-characterized roles of circRNAs is their ability to act as sponges for microRNAs (miRNAs) [74]. CircRNAs can prevent these small RNA molecules from interacting with their target mRNAs, effectively increasing the expression of those target genes by binding to miRNAs. CircRNAs can also interact with RNA-binding proteins and influence their activity, thereby affecting the expression of associated genes. CDR1as (also known as ciRS-7) is a well-studied circRNA that sequesters miR-7, a microRNA known to have tumor-suppressive functions [75]. CDR1as prevents it from regulating its target genes, which can lead to the increased expression of genes that promote LC cell proliferation and invasion by binding to miR-7 [76]. This mechanism highlights how circRNAs can contribute to the progression of cancer by modulating the availability of miRNAs that would otherwise inhibit tumor growth [77]. Both lncRNAs and circRNAs play significant roles in the regulation of gene expression and cellular processes in LC. Their ability to interact with various molecular players in the cell makes them critical components in the complex network of cancer biology.

## 3. ncRNAs as Diagnostic Biomarkers

The unique expression profiles of ncRNAs in LC tissues and their presence in body fluids such as blood and sputum highlight their potential as non-invasive diagnostic biomarkers. lncRNAs exhibit tissue-specific expression patterns, enhancing the specificity of cancer diagnostics. Certain lncRNAs are significantly upregulated or downregulated in LC tissues compared to normal tissues, as mentioned in Table 1 [78]. This differential expression can serve as a potential indicator of malignancy, aiding in the diagnosis of LC. The stability of lncRNAs in body fluids such as blood and urine allows for non-invasive sampling methods. This characteristic is particularly beneficial for LC, where traditional biopsy methods can be invasive and risky [79]. Many lncRNAs are linked to specific tumor characteristics, such as metastasis and drug resistance. lncRNAs like MALAT1 and HOTAIR have been associated with aggressive tumor behavior and poor patient outcomes in non-small-cell lung cancer (NSCLC) [80]. Their expression levels correlate with pathological stages and patient longevity, making them valuable for predictive assessments. lncRNAs are involved in various biological processes, including gene regulation and signaling pathways. Various lncRNAs can influence the MAPK signaling pathway, which is crucial in LC progression. The ability to detect lncRNAs in non-invasive samples reduces patient discomfort and risk. Advances in technologies such as RNA sequencing allow for the high-throughput detection of lncRNAs. The unique expression profiles of lncRNAs can aid in developing personalized treatment strategies.

### 3.1. miRNAs as Diagnostic Biomarkers

miRNAs have been extensively studied for their diagnostic potential. A panel of miRNAs, including miR-21, miR-210, and miR-155, has been identified in the serum of LC patients, demonstrating high sensitivity and specificity for early detection. miRNAs have emerged as promising diagnostic biomarkers for LC due to their stability in bodily fluids and their ability to regulate gene expression [93]. Several studies have identified specific miRNA signatures that correlate with LC presence, stage, and prognosis. miRNAs such as miR-21, miR-210, and miR-155 are significantly upregulated in the serum of LC patients [94]. These miRNAs can serve as non-invasive biomarkers, allowing for the early detection of LC through blood tests. Their expression levels can provide insights into the presence of malignancy, making them valuable for screening and diagnosis [95]. The identified miRNA panels have shown high sensitivity and specificity, meaning they can accurately distinguish LC patients from healthy individuals or those with benign conditions. This is crucial for early detection, as it can lead to timely intervention and improved patient outcomes [96]. miRNAs play a role in various cellular processes, including proliferation, apoptosis, and metastasis. miRNAs can influence tumor behavior by regulating target genes involved in these processes. miR-21 is known to promote cell survival and proliferation, while miR-155 is associated with inflammation and the immune response, both of which can contribute to cancer progression. The use of miRNA signatures in clinical practice could enhance current diagnostic protocols. They can complement existing imaging techniques and tissue biopsies, providing a more comprehensive approach to LC diagnosis. Furthermore, miRNA profiling may help in stratifying patients for personalized treatment plans based on their specific miRNA expression patterns.

### 3.2. lncRNAs as Diagnostic Biomarkers

Several lncRNAs have shown promise as diagnostic biomarkers. The lncRNA HOTAIR is upregulated in LC tissues, and its elevated levels in plasma correlate with disease stage and metastasis. HOTAIR has been found to be upregulated in NSCLC and is associated with tumor invasion and metastasis [97]. Its elevated levels in plasma can serve as a potential biomarker for diagnosing and monitoring NSCLC patients. This lncRNA is linked to poor prognosis in NSCLC and promotes tumor growth and metastasis. Studies have shown that MALAT1 can be a reliable biomarker for assessing the progression of LC [98]. Also, GAS5 has been identified as a novel biomarker for diagnosing NSCLC. Its circulating levels in plasma have been correlated with the presence of the disease, indicating its potential utility in clinical settings. This lncRNA has been detected in plasma and is considered a novel biomarker for esophageal squamous cell carcinoma [99]. Its presence in circulation reflects the pathological changes associated with cancer. HULC has been identified as a promising biomarker for hepatocellular carcinoma. Its levels in plasma can help in the early diagnosis of liver cancer, showcasing the potential of lncRNAs in cancer diagnostics [100].

### 3.3. circRNAs as Diagnostic Biomarkers

They are involved in various biological processes, including gene regulation, and have gained attention for their potential roles in cancer biology, particularly LC. CircRNAs are known for their stability in biological fluids, making them suitable candidates for biomarkers [101]. Their unique structure allows them to resist degradation, which is advantageous for diagnostic purposes. Research has shown that certain circRNAs are differentially expressed in LC tissues compared to normal tissues. Studies have identified specific circRNAs, such as hsa_circ_0077837 and hsa_circ_0001821, which demonstrate significant diagnostic value, with the AUC (Area Under the Curve) indicating their effectiveness in distinguishing non-small-cell LC from normal tissues [102]. CircRNAs can be detected in blood samples, offering a non-invasive method for early diagnosis. This is particularly important, as traditional biopsy methods can be invasive and carry risks [103]. Liquid biopsies that analyze circRNAs in plasma have shown promising results, suggesting their potential as emerging diagnostic markers for LC [104]. CircRNAs can be isolated from plasma exosomes, providing a less invasive alternative to tissue biopsies. Studies have identified specific circRNAs that are significantly upregulated in the early stages of lung adenocarcinoma, indicating their potential as early diagnostic markers. A meta-analysis of studies on circRNAs in LC has shown that they can achieve a pooled AUC of 0.78, suggesting a reasonable diagnostic potential in the Chinese LC population [105]. This indicates that circRNAs could be used to improve early detection strategies. CircRNAs can act as microRNA sponges, sequestering microRNAs and preventing them from binding to their target mRNAs. This regulatory mechanism can influence the expression of genes involved in cancer progression, such as those related to proliferation, metastasis, and apoptosis [106]. CircRNAs can function as either tumor-promoting or tumor-suppressing factors, depending on their specific roles in cellular pathways. They can regulate the biological behaviors of LC cells, impacting their growth and response to therapies.

## 4. ncRNAs as Prognostic Biomarkers

Non-coding RNAs (ncRNAs) play a significant role not only as diagnostic markers but as prognostic biomarkers in cancer, providing insights into disease outcomes and therapeutic responses. Certain ncRNAs, particularly microRNAs (miRNAs), have been associated with tumor aggressiveness. Elevated levels of miR-21 have been linked to poor outcomes across various cancer types, indicating its potential as a prognostic marker [107]. Long non-coding RNAs (lncRNAs) such as HOTAIR are involved in cancer metastasis and have been identified as survival predictors [108]. Their expression levels can help determine the likelihood of cancer spreading, which is crucial for patient management and treatment planning. ncRNAs can also indicate the likelihood of chemotherapy resistance [109]. Specific miRNA profiles have been correlated with responses to chemotherapy, allowing for the tailoring of treatment strategies based on individual patient profiles [110]. The stability of ncRNAs in bodily fluids makes them suitable for use in liquid biopsies. Clinicians can monitor changes over time, providing real-time insights into tumor dynamics and treatment responses by analyzing circulating ncRNAs, as mentioned in Table 2. This non-invasive approach enhances the ability to predict disease recurrence and guide adjuvant therapy decisions.

### miRNAs as Prognostic Biomarkers

miRNAs are small non-coding RNAs that play crucial roles in the regulation of gene expression. miR-200c is part of the miR-200 family, known to be involved in the regulation of the EMT, a process critical for cancer metastasis. miR-200c is frequently downregulated in LC, which contributes to tumor progression and metastasis. Its low expression has been correlated with aggressive tumor characteristics and poor patient outcomes. Low levels of miR-200c expression are associated with poor prognosis in LC patients. Patients with reduced miR-200c levels tend to have a lower overall survival rate and a higher chance of cancer recurrence. miR-200c downregulation is also linked to resistance to chemotherapy. This resistance makes treatment more challenging, as conventional chemotherapy becomes less effective in patients with low miR-200c levels. The primary mechanism by which miR-200c exerts its effects involves the inhibition of the EMT. The EMT is a process by which epithelial cells acquire mesenchymal properties, leading to enhanced migratory capacity, invasiveness, and resistance to apoptosis. miR-200c helps maintain the epithelial phenotype of cells and inhibits metastasis by suppressing the EMT. The EMT is a crucial biological process during which epithelial cells lose their cell–cell adhesion and gain mesenchymal properties. This transition enhances their migratory capacity, invasiveness, and resistance to apoptosis, facilitating cancer progression and metastasis. The EMT is characterized by the downregulation of epithelial markers (e.g., E-cadherin) and the upregulation of mesenchymal markers (e.g., N-cadherin, Vimentin).

By inhibiting ZEB1 and ZEB2, miR-200c ensures the continued expression of E-cadherin, a key epithelial marker. E-cadherin is crucial for maintaining tight junctions and cell polarity, which are characteristic of epithelial cells. miR-200c helps maintain the organization of the actin cytoskeleton, which is essential for the structural integrity and function of epithelial cells. Restoring miR-200c levels in LC cells presents a potential therapeutic strategy. It may be possible to reduce tumor aggressiveness, improve the effectiveness of chemotherapy, and ultimately enhance patient outcomes by increasing miR-200c expression. miR-200c downregulates the expression of mesenchymal markers such as N-cadherin and Vimentin, which are associated with increased cell motility and invasiveness [123]. Moreover, miR-200c reduces the invasive and migratory capabilities of cancer cells by suppressing the expression of proteins involved in cell motility. The EMT is associated with resistance to apoptosis, which allows cancer cells to survive under adverse conditions. miR-200c promotes apoptosis by inhibiting the EMT, thereby increasing the susceptibility of cancer cells to cell death [124]. miR-200c targets survival pathways that are activated during the EMT, further contributing to the induction of apoptosis in cancer cells [125].

## 5. Therapeutic Response Monitoring

### 5.1. ncRNAs

Monitoring ncRNA expression can also provide insights into therapeutic responses. ncRNAs are a class of RNAs that do not encode proteins but play significant roles in regulating gene expression and cellular processes. Their involvement in cancer has been increasingly recognized, making them valuable for cancer diagnosis and prognosis and as therapeutic targets. ncRNAs, including miRNAs, lncRNAs, and circRNAs, serve as biomarkers for the early detection of cancers [126]. Their expression profiles can differentiate between cancerous and non-cancerous tissues. ncRNAs in bodily fluids like blood and urine enable non-invasive cancer diagnostics [127]. Techniques like liquid biopsies use ncRNA signatures to detect and monitor cancer. ncRNA levels can provide prognostic information about the aggressiveness of the cancer, potential for metastasis, and overall patient survival [128]. Low levels of certain miRNAs are associated with poor prognosis in LC patients [129]. The expression of specific ncRNAs can predict the response to chemotherapy and other treatments, aiding in personalized treatment plans. ncRNAs can be targeted to modulate their expression, offering a novel approach to cancer therapy [130]. Restoring tumor-suppressive ncRNAs or inhibiting oncogenic ncRNAs can inhibit cancer progression. ncRNAs are being explored as targets for drug development. Therapies that mimic or inhibit ncRNAs are in various stages of research and clinical trials [131]. ncRNAs like miR-21 and lncRNA HOTAIR are involved in breast cancer progression and are being studied as therapeutic targets [132]. circRNA_0000190 and miR-200c have been identified as critical players in LC, with implications for prognosis and treatment resistance (illustrated in Figure 3). ncRNAs like PCA3 and miR-141 are used as biomarkers for early detection and prognosis of prostate cancer [133]. Standardizing ncRNA detection methods and establishing robust clinical protocols are essential for their widespread adoption. There are various techniques for detecting ncRNAs, including qRT-PCR, microarrays, and next-generation sequencing (NGS) [134]. Further clinical trials are needed to validate ncRNA-based therapies and their effectiveness in different cancer types, tailoring ncRNA-based treatments to individual patient profiles based on their unique ncRNA expression patterns. Combining miR-34 mimics with EGFR-TKIs could potentially overcome resistance and improve outcomes in LC patients [135].

### 5.2. lncRNAs

Recent advancements in the molecular understanding of LC have highlighted the significance of lncRNAs as potential biomarkers for diagnosis, prognosis, and therapeutic targeting. lncRNA PVT1 (Plasmacytoma Variant Translocation 1) has emerged as a critical player in the pathology of LC [136]. lncRNA PVT1 is a multifunctional RNA molecule located at chromosomal band 8q24, a region frequently associated with various malignancies. In the context of LC, numerous studies have demonstrated a consistent upregulation of PVT1 across different subtypes of the disease, including NSCLC and small-cell lung cancer (SCLC). Elevated expression levels of PVT1 have been linked to aggressive tumor behavior, increased proliferation, and enhanced metastatic potential [137]. Mechanistically, lncRNA PVT1 exerts its oncogenic effects through several pathways. It is known to interact with key regulatory proteins and miRNAs, influencing cell cycle progression, apoptosis, and the EMT [138]. PVT1 has been shown to act as a molecular sponge for tumor-suppressive miRNAs, such as miR-195 and miR-29, thereby disrupting their tumor-suppressive functions. Additionally, PVT1′s interaction with chromatin-modifying complexes can affect the transcriptional landscape of cancer cells, promoting an oncogenic phenotype [139]. The prognostic value of lncRNA PVT1 in LC has been increasingly substantiated by clinical studies. Elevated PVT1 expression levels have been statistically associated with poor overall survival (OS) and progression-free survival (PFS) in LC patients. A high PVT1 expression correlates with an advanced tumor stage, an increased likelihood of lymph node metastasis, and a reduced response to conventional therapies [140]. These observations underscore PVT1′s potential utility as a prognostic biomarker. Kaplan–Meier survival analyses and multivariate Cox proportional hazards models have consistently demonstrated that patients with high PVT1 expression exhibit significantly shorter survival times compared to those with lower levels of PVT1 [141]. Furthermore, the integration of PVT1 expression profiling into clinical practice could refine risk stratification and enable personalized therapeutic strategies, enhancing treatment efficacy and patient outcomes. Despite the promising evidence, further research is required to fully elucidate the functional mechanisms of lncRNA PVT1 and to validate its clinical utility [142]. Large-scale, multicentric studies are needed to confirm the reproducibility of PVT1′s prognostic value across diverse populations and treatment regimens [143]. Additionally, the development of targeted therapies aimed at modulating PVT1 activity represents a potential avenue for novel therapeutic interventions [144].

lncRNA UCA1 expression levels correlate with the response to EGFR-TKI therapy in NSCLC patients [145]. NSCLC is one of the most common and deadly forms of LC [146]. Epidermal growth factor receptor tyrosine kinase inhibitors (EGFR-TKIs) are a targeted therapy used to treat NSCLC patients with mutations in the EGFR gene [147]. However, the response to EGFR-TKI therapy varies among patients, necessitating the identification of biomarkers to predict treatment outcomes [148]. lncRNA urothelial carcinoma-associated 1 (UCA1) is a well-studied lncRNA known for its involvement in various cancers, including NSCLC. Research has shown that the expression levels of lncRNA UCA1 can influence the response to EGFR-TKI therapy in NSCLC patients. It is well known that the EGFR-dependent resistance mechanisms mainly refer to EGFR site mutations, among which T790M, a secondary mutation in the EGFR site, is found to be the leading cause for resistance to the first- and second-generation EGFR-TKIs, such as gefitinib and erlotinib [149]. The EGFR C797S tertiary mutation is the most frequent EGFR-dependent resistance mechanism of third-generation EGFR-TKIs such as Osimertinib [150]. There is not enough proof to support the role of lncRNAs in EGFR-dependent resistance mechanisms [151]. However, numerous investigations have shown that lncRNAs have broad control over EGFR-independent resistance mechanisms in NSCLC. Extensive investigations have documented that the upstream receptors, regulatory factors, and downstream molecules can be regulated by lncRNAs to modulate EGFR-TK1 resistance in NSCLC. Even at the moment, multiple studies are being conducted on the tumor microenvironment of LC that focus on immune cells and immunotherapy. Substantial evidence also indicates that lncRNAs in tumor cells or other cells can be transported into tumor microenvironments to regulate cell function and promote drug resistance [152].

Elevated levels of UCA1 have been associated with resistance to EGFR-TKI therapy. Conversely, lower UCA1 expression is often correlated with a better response to this treatment [78]. Thus, UCA1 expression can serve as a predictive biomarker for determining which patients are more likely to benefit from EGFR-TKIs. UCA1 is involved in several cellular processes, including proliferation, apoptosis, and metastasis. In the context of EGFR-TKI therapy, UCA1 may regulate signaling pathways that affect the sensitivity of cancer cells to the treatment. UCA1 can modulate the PI3K/AKT pathway, which is crucial for cell survival and can contribute to drug resistance [153]. Measuring UCA1 levels in NSCLC patients before initiating EGFR-TKI therapy can help oncologists personalize treatment plans. Patients with high UCA1 expression may be considered for alternative therapies or combination treatments to overcome resistance [154]. Studies have demonstrated that silencing UCA1 in NSCLC cell lines increases their sensitivity to EGFR-TKIs, while the overexpression of UCA1 leads to a reduced efficacy of these drugs [155]. These findings highlight the potential of targeting UCA1 to enhance the effectiveness of EGFR-TKI therapy. Developing drugs that specifically inhibit UCA1 could improve the response to EGFR-TKIs in NSCLC patients. These targeted therapies could be used in combination with EGFR-TKIs to overcome resistance mechanisms. Incorporating UCA1 expression analysis into diagnostic tests for NSCLC could aid in the stratification of patients based on their likelihood of responding to EGFR-TKI therapy [156]. This approach would enable more tailored and effective treatment strategies. lncRNA UCA1 is a promising biomarker for predicting the response to EGFR-TKI therapy in NSCLC patients [157]. Understanding the expression levels of UCA1 can help guide treatment decisions, improve patient outcomes, and pave the way for new therapeutic interventions targeting this lncRNA.

### 5.3. miRNAs

The levels of miR-125b and miR-221 have been associated with the response to chemotherapy in LC patients, particularly in NSCLC [158]. This microRNA is often downregulated in various cancers, including NSCLC, where it acts as a tumor suppressor [25]. Its downregulation is linked to increased tumor proliferation and resistance to chemotherapy. Studies have shown that lower levels of miR-125b in tumor tissues correlate with poorer responses to chemotherapy agents like cisplatin [159]. This suggests that monitoring miR-125b levels could provide insights into the effectiveness of treatment and help predict patient outcomes. Conversely, miR-221 is typically upregulated in many cancers, including LC, and is associated with poor prognosis and increased resistance to chemotherapy [160]. Elevated levels of miR-221 can promote cell survival and proliferation by targeting pro-apoptotic genes, thereby contributing to drug resistance. Therefore, high levels of miR-221 may indicate a reduced efficacy of chemotherapy in LC patients. miR-125b is often considered a tumor suppressor in various cancers [161]. It regulates several target genes involved in cell proliferation, apoptosis, and differentiation. It can downregulate anti-apoptotic proteins like Bcl2, promoting apoptosis in cancer cells [162]. miR-125b influences critical signaling pathways such as the NF-κB pathway, which is involved in inflammation and cell survival [163]. miR-125b can enhance cell proliferation and survival, contributing to tumor growth in certain contexts by suppressing A20, a negative regulator of NF-κB [164]. In some cancers, miR-125b has been implicated in promoting invasion and metastasis. Its downregulation has been linked to increased invasion in endometrial cancer by targeting ERBB2, a gene associated with cell signaling and growth. miR-125b can also play a role in drug resistance [165]. In certain cancer types, its expression levels can influence the sensitivity of tumors to chemotherapy agents. The downregulation of miR-125b has been associated with resistance to cisplatin in gallbladder cancer [166].

The expression levels of miR-125b can serve as biomarkers for cancer diagnosis and prognosis. Low levels of miR-125b in tumor tissues may indicate a more aggressive disease and poorer outcomes, while higher levels could suggest a better prognosis in some contexts [167,168]. Given its dual role in different cancers, miR-125b could be a potential therapeutic target [169]. Strategies to restore or mimic miR-125b function in tumors where it is downregulated could enhance the effectiveness of existing therapies and reduce tumor growth [170]. Changes in miR-125b levels during treatment could be used to monitor the efficacy of chemotherapy. An increase in miR-125b levels in response to treatment may indicate a positive therapeutic effect, while a decrease could suggest resistance. miR-125b could be integrated into combination therapy strategies [171]. Combining miR-125b mimics with conventional chemotherapy might enhance treatment efficacy by overcoming drug resistance mechanisms

## 6. Conclusions

Non-coding RNAs hold great promise as biomarkers for LC, offering potential for early diagnosis, prognosis, and therapeutic response monitoring. Advances in ncRNA research have paved the way for developing novel diagnostic tools and targeted therapies, which could significantly improve patient outcomes [172]. Future research should focus on validating ncRNA biomarkers in large clinical cohorts and understanding the mechanisms underlying their roles in LC to fully harness their clinical potential [173]. The identification and validation of specific ncRNA signatures hold promise for revolutionizing early diagnosis. ncRNAs can be detected in various biofluids, including blood and sputum, offering minimally invasive diagnostic options. Circulating miRNAs such as miR-21 and miR-155 have shown potential as biomarkers for the early detection and stratification of LC [174]. The specificity and sensitivity of these biomarkers could be further refined through high-throughput profiling and integration with other diagnostic modalities. In terms of prognosis, ncRNAs provide valuable information beyond traditional histopathological and clinical parameters. High expression levels of certain lncRNAs, such as PVT1, have been associated with poor survival outcomes and resistance to therapy, as noted in the referenced article [34]. These findings underscore the need for comprehensive ncRNA profiling to develop prognostic models that predict disease progression and patient survival more accurately [175]. Monitoring therapeutic responses through ncRNA expression patterns offers a dynamic approach to assessing treatment efficacy and guiding therapeutic adjustments. Changes in the levels of specific miRNAs or lncRNAs in response to treatment can provide real-time insights into therapeutic effectiveness and resistance mechanisms [176]. Such monitoring can facilitate personalized treatment regimens and improve patient outcomes by enabling timely interventions based on individual molecular responses [177]. The rigorous validation of ncRNA biomarkers in large, diverse patient populations is essential to establish their clinical utility and reliability [178]. Multicentric studies and meta-analyses will help validate findings across different ethnic and clinical settings, ensuring the robustness and generalizability of ncRNA-based biomarkers. Detailed studies are needed to elucidate the precise mechanisms by which ncRNAs influence LC biology [179]. Understanding the interactions between ncRNAs and their target genes, as well as their roles in key processes such as metastasis and drug resistance, will provide deeper insights into their functional significance. Translating ncRNA research into clinical practice involves developing practical assays for routine use, establishing guidelines for biomarker-based decision-making, and ensuring the integration of these biomarkers into existing diagnostic and therapeutic frameworks [180]. Beyond diagnostics, ncRNAs present opportunities for novel therapeutic interventions. Exploring ncRNA-based therapies, such as antisense oligonucleotides or small molecules targeting specific ncRNAs, could lead to innovative treatment options that directly modulate cancer-related ncRNA pathways [181]. In conclusion, the advancement of ncRNA research offers substantial promise for improving LC management [182].

## Figures and Tables

**Figure 1 ncrna-10-00050-f001:**
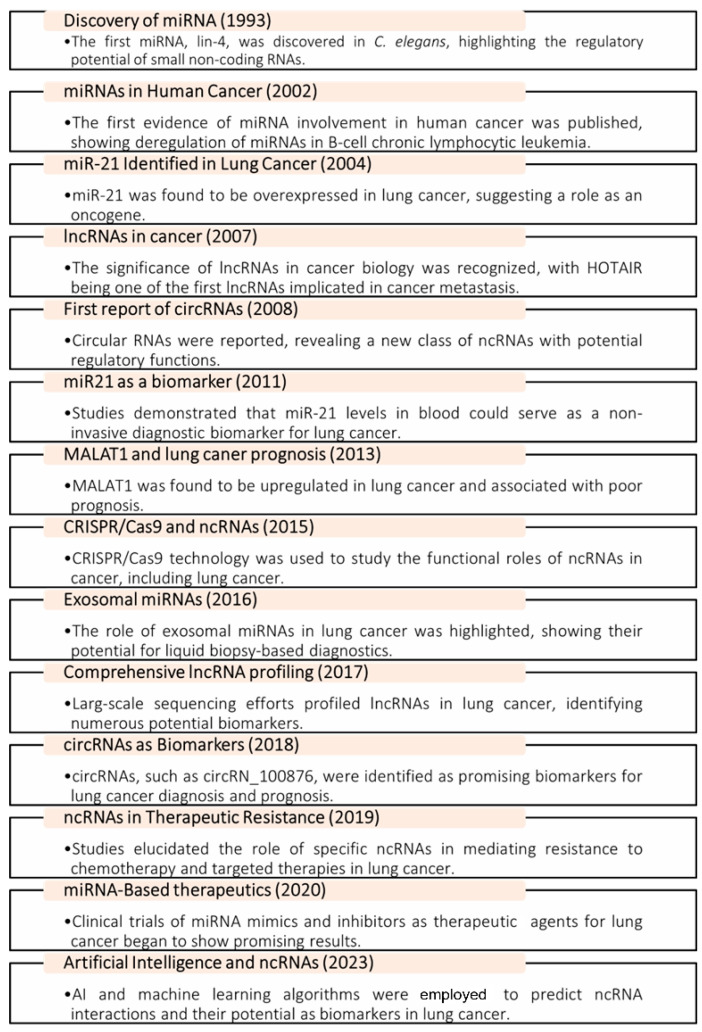
Timeline of milestones in ncRNA research in lung cancer.

**Figure 2 ncrna-10-00050-f002:**
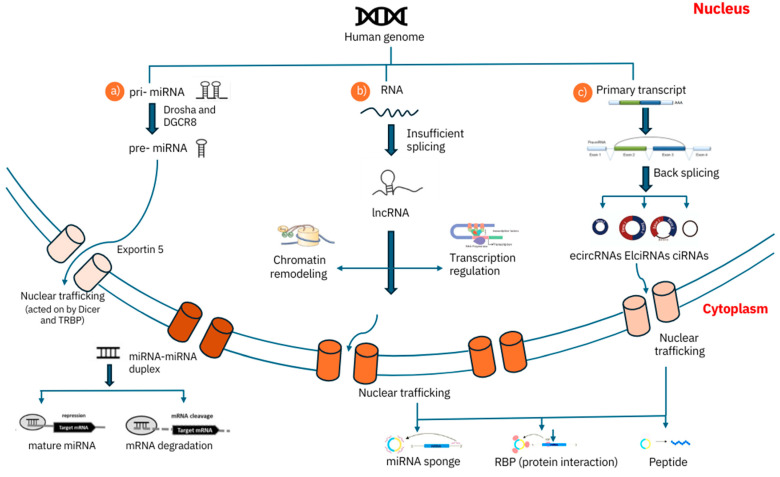
Biogenesis and function of ncRNAs in lung cancer: Biogenesis and Function of ncRNAs in Lung Cancer. This schematic illustrates the biogenesis pathways of miRNAs, lncRNAs, and circRNAs, and their diverse roles in lung cancer biology. Key functions depicted include gene regulation through various mechanisms, chromatin remodeling, and miRNA sponging. The figure highlights the complex interplay between different types of ncRNAs and their impact on cellular processes relevant to lung cancer development and progression.

**Figure 3 ncrna-10-00050-f003:**
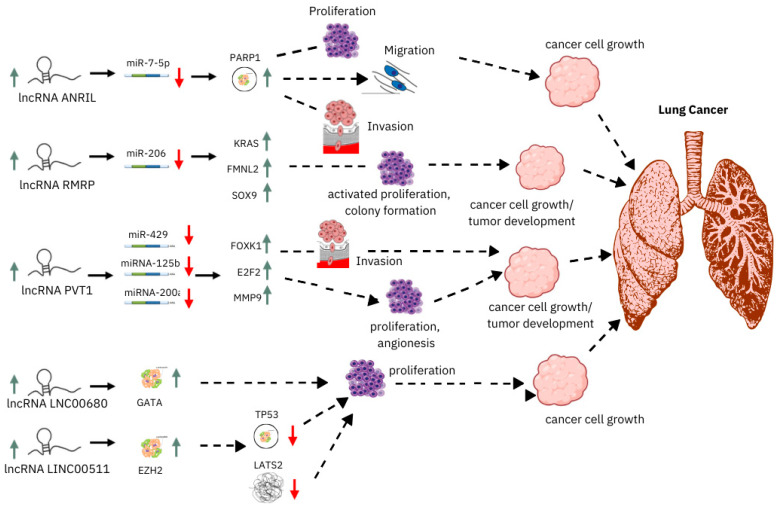
Role of lncRNAs in lung cancer pathogenesis. This schematic effectively conveys the multifaceted involvement of ncRNAs in lung cancer pathogenesis. A visual demonstration of how different ncRNAs, including lncRNA APRIL, lncRNA RMRP, lncRNA PVT1, LNC00680, LINC00511 influence critical processes that contribute to cancer development, progression, and treatment challenges.

**Table 1 ncrna-10-00050-t001:** Summary of diagnostic ncRNA biomarkers in lung cancer.

ncRNA	Type	Expression in Lung Cancer	Detection Method	Diagnostic Accuracy	Source	Sample Size/Method	References
miR-21	miRNA	Upregulated	qRT-PCR	High sensitivity and specificity	Tissue and A549 cells	Meta-analyses use quantitative reverse-transcription PCR (qRT-PCR)	[81]
HOTAIR	lncRNA	Upregulated	qRT-PCR	Correlates with disease stage	Tissue	818 NSCLC patients vs. adjacent normal lung tissue samples; used quantitative real-time PCR (qRT-PCR)	[82]
circRNA_100876	circRNA	Upregulated	qRT-PCR	Differential expression in plasma	Tissue	101 pairs of NSCLC tissues compared to adjacent normal lung tissues; used quantitative real-time PCR (qRT-PCR)	[83]
MALAT1	lncRNA	Upregulated	qRT-PCR	Area under the ROC curve (AUC) of 0.79	Tumor tissues, blood/plasma, and urine	Quantitative real-time PCR (qRT-PCR), RNA sequencing, and in situ hybridization	[84]
SPRY4-IT1, ANRIL, and NEAT1	lncRNA	Upregulated	qRT-PCR	AUC of 0.876; high sensitivity and specificity in NSCLC	Plasma	Detected in 50 patients and 50 volunteers using quantitative real-time PCR	[84]
GAS5	lncRNA	Upregulated	qRT-PCR	AUC of 0.832; differential expression in NSCLC tissues and plasma	Tissue samples and liquid biopsies (blood/plasma)	Detected in 40 patients with advanced stage (III and IVB) lung adenocarcinoma using qRT-PCR	[84]
LINC00963 and DLX6-AS1	lncRNA	Upregulated	qRT-PCR	Differential expression in plasma associated with lung adenocarcinoma (LUAD)	Plasma samples	90 lung cancer patients + 90 healthy controls, using qRT-PCR	[85]
ADAMTS9-AS2	lncRNA	Upregulated	qRT-PCR	Differential expression in plasma	Blood/plasma	Quantitative real-time PCR (qRT-PCR), RNA sequencing (RNA-seq), and microarray analysis	[86]
XLOC_009167	lncRNA	Upregulated	qRT-PCR	Differentially expressed lncRNAs between tumor and normal tissues were identified	Serum	Using quantitative real-time PCR (qRT-PCR) data from the GEO database	[87]
miRNA-192	miRNA	Upregulated	qRT-PCR	Contributes to the early diagnosis of NSCLC	Tissue samples, serum, and plasma	1038 cancer patients and 938 healthy controls	[88]
miRNA-17	miRNA	Upregulated	qRT-PCR	Diagnostic biomarker for lung adenocarcinoma screening	Serum	5 microarray datasets from the Gene Expression Omnibus (GEO) database, comprising a total of 87 LUAD samples and 83 healthy controls, using quantitative real-time PCR (qRT-PCR) data	[89]
miR-25	miRNA	Upregulated	qRT-PCR	Enhanced diagnostic accuracy for NSCLC in liquid biopsy settings	Serum or plasma samples	6 studies with a total of 480 NSCLC patients and 451 healthy controls, using qRT-PCR	[90]
RMRP	lncRNA	Upregulated	qRT-PCR	Upregulated in CRC tissues as compared to adjacent normal tissues	Serum, plasma, and tissue	Quantitative real-time PCR (qRT-PCR), RNA sequencing (RNA-seq), and in situ hybridization	[91]
NEAT1	miRNA sponge	Upregulated	qRT-PCR	Part of a 4-lncRNA panel with high diagnostic value in NSCLC	Tissue	208 lung cancer samples, 208 non-cancer samples, quantitative real-time PCR (qRT-PCR), RNA sequencing (RNA-seq), in situ hybridization (ISH), and fluorescence in situ hybridization (FISH)	[92]

**Table 2 ncrna-10-00050-t002:** Prognostic ncRNA biomarkers in lung cancer.

ncRNA	Type	Expression in Lung Cancer	Prognostic Implication	Source	Sample Size/ Detection Method	Reference
miR-200c	miRNA	Downregulated	Poor prognosis and chemo-resistance	Tissue and A549 cells	Meta-analyses use quantitative reverse-transcription PCR (qRT-PCR)	[111]
PVT1	lncRNA	Upregulated	Poor survival rates	Tissue and cell lines	Quantitative real-time PCR (qRT-PCR), RNA sequencing (RNA-seq), and in situ hybridization (ISH)	[112]
circRNA_0000190	circRNA	Upregulated	Tumor progression and poor prognosis	NSCLC tissues and cell lines	Quantitative real-time PCR (qRT-PCR), RNA sequencing (RNA-seq), and fluorescence in situ hybridization (FISH)	[113]
AC099850.3	lncRNA	Upregulated	Overall survival (OS), disease-free survival (DSS), and progress-free survival (PFS)	Tissue and cell lines including COLO 320 and SK-PN-DW	Quantitative real-time PCR (qRT-PCR), RNA sequencing (RNA-seq), and in situ hybridization (ISH)	[114]
DPP10-AS1	lncRNA	Upregulated	Promotes proliferation; associated with poor prognosis	Tissues and cell lines	94 paired lung cancer tissues and adjacent normal tissues, detected using quantitative real-time PCR (qRT-PCR) and in situ hybridization (ISH)	[115]
KTN1-AS1	lncRNA	Upregulated	Correlated with TNM stage, histological grade, and lymph node metastasis; high expression reduces OS	NSCLC tissues and cell lines	90 pairs of NSCLC tissues and adjacent normal tissues, using quantitative real-time PCR (qRT-PCR)	[116]
PTTG3P	lncRNA	Upregulated	High expression associated with shorter OS and DFS in NSCLC patients	Cell lines including A549, H1299, PC-9, 16HBE, etc.	Using quantitative real-time PCR (qRT-PCR) in these cell lines	[117]
ccdc144nl-AS1	lncRNA	Upregulated	Promotes cellular function by targeting miR-490-3p	NSCLC tissues and cell lines	128 pairs of NSCLC tissues and paracancerous tissues, using quantitative real-time PCR (qRT-PCR)	[118]
AC018629.1	lncRNA	Upregulated	Part of a four-lncRNA signature associated with overall survival in LUAD patients	Lung adenocarcinoma tissues using data from The Cancer Genome Atlas (TCGA) database	446 LUAD patients from TCGA database using RNA sequencing data from TCGA database	[85]
LINC01833	lncRNA	Upregulated	Correlates with immune infiltrates in LUAD patients	Lung adenocarcinoma tissues using bioinformatics analysis of public databases	Hundreds of LUAD samples were analyzed using RNA sequencing data from TCGA database	[119]
AL138789.1	lncRNA	Upregulated	Part of a four-lncRNA signature associated with overall survival in LUAD patients	Lung adenocarcinoma tissues using data from The Cancer Genome Atlas (TCGA) database	535 LUAD samples from TCGA database were analyzed using RNA sequencing data from TCGA database	[120]
AC119424.1	lncRNA	Upregulated	Part of a four-lncRNA signature associated with overall survival in LUAD patients	LUAD and normal tissues	535 LUAD samples and 59 normal lung samples; RNA sequencing data analysis from TCGA microarray data analysis from GEO datasets	[121]
AC122134.1	lncRNA	Upregulated	Part of a four-lncRNA signature associated with overall survival in LUAD patients	Lung adenocarcinoma tissues using data from The Cancer Genome Atlas (TCGA) database	535 LUAD samples and 59 normal lung tissue samples from TCGA database, detected using RNA sequencing data from TCGA database	[85]
circ_0001946	circRNA	Upregulated	Promotes tumor progression by sponging miR-135a-5p	Tissues and cell lines	RNA sequencing (RNA-seq), quantitative real-time PCR (qRT-PCR), and microarray analysis	[122]

## Data Availability

No new data were created or analyzed in this study. Data sharing is not applicable to this article.

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
