# Peer review of "Non-Coding RNA as a Biomarker in Lung Cancer"

_ncrna, 2024, doi:10.3390/ncrna10050050_

Round 1

Reviewer 1 Report

Comments and Suggestions for Authors

Overall, review is nicely written but lacks some of the essentials informations in tables.

Some of suggestions to improve the article are as follows-

1. Numbers of the patients/samples included in cited studies are essential together with detection methods and other criteria in the table 1 and 2.

2. Are there studies with other detection methods than qRT-PCR such as RNA sequencing or microarrays? If, yes please include those studies too with cohort sizes. 

3. Current incident and mortality rate should be mentioned in introduction part.

4. Circular RNAs could be highlighted separately instead of in long non-coding RNAs section. 

Comments on the Quality of English Language

The presented form of review article need to be revised to improve some of the grammatical and typos mistakes throughout the manuscript. 

Author Response

Comments and Suggestions for Authors

Rivewer1

Overall, review is nicely written but lacks some of the essentials informations in tables.

Some of suggestions to improve the article are as follows-

  1. Numbers of the patients/samples included in cited studies are essential together with detection methods and other criteria in the table 1 and 2.

Answer: Thank you for your insightful comment regarding the inclusion of patient/sample numbers, detection methods, and other criteria in Tables 1 and 2. We fully agree that these details are essential for the proper interpretation of the data presented in our manuscript. As per the suggestion we incorporated in the manuscript

  1. Are there studies with other detection methods than qRT-PCR such as RNA sequencing or microarrays? If, yes please include those studies too with cohort sizes. 

We mentioned this in the above table.

Answer: Thank you for your valuable feedback regarding the detection methods used in the studies cited in our manuscript. In response to your query, we have conducted a thorough review of the literature to identify studies that employed detection methods other than qRT-PCR, such as RNA sequencing or microarrays and included in Table.

  1. Current incident and mortality rate should be mentioned in introduction part.

Answer: Thank you for your valuable suggestion. We have included a complete para in the introduction

4. Circular RNAs could be highlighted separately instead of in long non-coding RNAs section. 

Answer: Thank you for your valuable suggestion. We have included separate heading in the manuscript

Thank you very much

Best regards

Dr Henu Kumar Verma

Reviewer 2 Report

Comments and Suggestions for Authors

This comprehensive review explores the emerging role of non-coding RNAs in lung cancer diagnostics and prognosis. The authors focused on various types of ncRNAs, including microRNAs, long non-coding RNAs, and circular RNAs and discussed their potential as novel biomarkers. They also highlighted the challenges and future directions in translating these findings into clinical applications for the early detection and personalized treatment of lung cancer. This review brings together a wide body of information that may be useful to researchers interested in this field. My specific comments on this manuscript are as follows:

Major points:

1.      The authors have forgotten to mention some ncRNAs that are already known as lung cancer biomarkers, such as circFARSA and circFoxo3.

2.      The authors should refer to the original manuscript, not a review article. For example, they should cite “Zhang L, Mol. Biotechnol. 2021”, not “Int J Bio Macromol, 2022” for ccdc144nl-AS1.

3.      Tables 1 and 2: Please show the histological types (e.g., LUAD) and sources (e.g., tissue or plasma) of each ncRNA reported.

Minor points:

1.      References: Authors should adjust their reference style, e.g., 24, 50, 51, 52, 53.

2.      Figure 1: They should reveal when the first report of circRNAs was published. Is that 2008 or 2011?

Author Response

Reviewer 2

Comments and Suggestions for Authors

This comprehensive review explores the emerging role of non-coding RNAs in lung cancer diagnostics and prognosis. The authors focused on various types of ncRNAs, including microRNAs, long non-coding RNAs, and circular RNAs and discussed their potential as novel biomarkers. They also highlighted the challenges and future directions in translating these findings into clinical applications for the early detection and personalized treatment of lung cancer. This review brings together a wide body of information that may be useful to researchers interested in this field. My specific comments on this manuscript are as follows:

Major points:

  1. The authors have forgotten to mention some ncRNAs that are already known as lung cancer biomarkers, such as circFARSA and circFoxo3.

Answer: Thank you for your valuable suggestion. Based on your suggestion, we have added the following paragraph to the paper.

CircFARSA has been found to be significantly elevated in the plasma of patients with non-small cell lung cancer (NSCLC)[5]. Its increased expression is associated with tumor aggressiveness and poor prognosis. CircFARSA shows potential as a non-invasive diagnostic and prognostic biomarker for NSCLC, allowing for early detection and monitoring of disease progression through simple blood tests. CircFoxo3, on the other hand, exhibits variable expression patterns across different cancer types, including lung cancer. In NSCLC specifically, circFoxo3 has been reported to be downregulated in tumor tissues and cell lines. It acts as a tumor suppressor by inhibiting cell proliferation, migration, and invasion through various molecular mechanisms, including sponging oncogenic microRNAs. The decreased expression of circFoxo3 in NSCLC patients' samples correlates with advanced tumor stage and poorer survival outcomes, highlighting its potential as a prognostic biomarker. Additionally, circFoxo3 levels in blood samples show promise for non-invasive early detection of lung cancer. Both circFARSA and circFoxo3 demonstrate the clinical utility of circular RNAs as stable, specific biomarkers that can be detected in liquid biopsies. Their altered expression patterns in lung cancer patients provide valuable diagnostic and prognostic information, potentially improving early detection rates and treatment strategies for this deadly disease.

  1. The authors should refer to the original manuscript, not a review article. For example, they should cite “Zhang L, Mol. Biotechnol. 2021”, not “Int J Bio Macromol, 2022” for ccdc144nl-AS1.

Answer: Thank you for your valuable suggestion. we have modified the reference

  1. Tables 1 and 2: Please show the histological types (e.g., LUAD) and sources (e.g., tissue or plasma) of each ncRNA reported.

Answer: Thank you for your valuable suggestion. The source is mentioned above for both tables. However, the histological types are not available for each study.

Minor points:

  1. References: Authors should adjust their reference style, e.g., 24, 50, 51, 52, 53.

Answer: Thank you for your valuable suggestion. we have updated the reference

  1. Figure 1: They should reveal when the first report of circRNAs was published. Is that 2008 or 2011?

Answer: Thank you for your valuable suggestion. We updated figure 1

Thank you very much

Best regards

Dr Henu Kumar Verma